# Barriers to Accessing Maternal Care in Low Income Countries in Africa: A Systematic Review

**DOI:** 10.3390/ijerph17124292

**Published:** 2020-06-16

**Authors:** Rana Dahab, Dikaios Sakellariou

**Affiliations:** 1Formerly London School of Hygiene and Tropical Medicine, University of London, London WC1E 7HT, UK; rana.i.dahab@gmail.com; 2School of Healthcare Sciences, Cardiff University, Cardiff CF24 0AB, UK

**Keywords:** access to maternal care, low-income countries, maternal mortality, maternal health, sustainable development goals

## Abstract

The new Sustainable Development Goals (SDGs) to 2030 aim to reduce maternal mortality and provide equitable access to maternal healthcare. Compromised access to maternal health facilities in low-income countries, and specifically in Africa, contribute to the increased prevalence of maternal mortality. We conducted a systematic review to investigate access barriers to maternal health in low-income countries in Africa since 2015, from the perspective of both community members and health providers. The findings show that the most important barriers to maternal health are transportation barriers to health facilities, economic factors, and cultural beliefs, in addition to lack of family support and poor quality of care. Further research is required to guide policymakers towards firm multi-sectoral action to ensure appropriate and equitable access to maternal health in line with the SDGs to 2030.

## 1. Introduction

Maternal health is a cornerstone for healthy and productive populations [1,2]. Although the United Nation’s (UN) Millennium Development Goals (MDGs) aimed to reduce global maternal mortality ratios by 75% by 2015 [3], there are still 830 women dying every day worldwide from preventable causes due to improper maternal care [4]. 99% of mortality cases take place in low-income countries (LICs), with more than half of these deaths occurring in Sub-Saharan Africa [5,6], where maternal mortality is still a persisting challenge.

Goal 3 of the new Sustainable Development Goals (SDGs) to 2030 aims to reduce global maternal mortality ratio to less than 70 per 100,000 live births [5,7,8,9]. Maternal mortality in several African countries is unpredictable yet preventable [7]. Maternal mortality could be attributed to poor socio-economic conditions, low quality of care (QoC), lack of well-trained healthcare professionals, lack of proper infrastructure, and barriers to accessing medical facilities [1,5,7,10]. Insecurity and scarce resources are additional critical issues to maternal healthcare accessibility for women living in conflict zones and fragile settings [11]. These factors could contribute to high risk of maternal bleeding, complications, and infections during childbirth and unsafe abortions [7,12]. Evidence suggests that reducing global maternal mortality and providing equitable access to healthcare can have many benefits for societies, including increased productivity and higher educational attainment [13,14,15,16,17,18].

To date, there has been insufficient evidence on the barriers to accessing maternal healthcare in LICs in Africa post the MDGs 2015 era. Most of the publications have either dealt with individual countries in Sub-Saharan Africa [7,8,19] or discussed general accessibility barriers in LICs [20]. Due to the failure to achieve the UN goal in 2015 to substantially reduce maternal mortality in Africa [3], more evidence-based work is required to further investigate the issue, particularly for LICs in Africa, which are more vulnerable to socio-economic barriers, low literacy levels, poorly resourced facilities, and poor transportation services, in addition to fragile country systems, all of which increase the risks of countries’ vulnerability to compromised access to maternal healthcare [1,21].

One of the underlying causes of high maternal mortality in LICs are barriers to accessing maternal health services [1,5,7,10]. Many influencing factors could affect the overall accessibility and utilization of maternal healthcare facilities (MHFs) [10,19]. Identifying the accessibility issues to maternal health in LICs in Africa could guide decision-makers’ to take the required measures towards appropriate and equitable healthcare, which can help transform the socio-economic status of LICs in Africa [22]. In light of the current efforts of the UN for the new SDGs to 2030 to provide equitable health access and reduce maternal mortality, this systematic review sought to offer an overview of perceived barriers to maternal healthcare in one of the most vulnerable settings, LICs in Africa. This review aimed to investigate barriers to accessing maternal care in LICs in Africa from the perspective of both women and relevant stakeholders, through the following objectives: investigate the experience of provision of maternal care services and facilities in LICs in Africa and explore the different barriers to accessing maternal care reported by women and stakeholders.

## 2. Materials and Methods

An extensive search was conducted in May 2019 for peer-reviewed articles through several electronic databases (Medline, Global Health, and CINAHL Plus), to ensure diversity of sources and reduced reporting and publication bias. The review protocol was not published.

### 2.1. Inclusion and Exclusion Criteria

Studies had to meet all of the following inclusion criteria: peer-reviewed publications on maternal healthcare access in LICs in Africa; women’s or healthcare workers’ (HCWs) perspectives on accessing MHFs; qualitative and mixed-method research; published work in English language; full-text publications, with a publication date between 2015 and May 2019, because we wanted to investigate access to maternal care following the end of MDGs in 2015.

Studies were excluded if they met one or more of the following criteria: publications addressing barriers to contraception; factors associated with home deliveries; and articles where data collection with mothers took place two years or more since birth.

### 2.2. Definitions

In this review, we used the World Health Organization’s definition of *maternal health* as women’s health during pregnancy, delivery, and the immediate period after childbirth [4]. *Maternal healthcare access* was defined as access related to utilization of maternal healthcare services, timely decision to seek care, physical accessibility to health facilities, and receiving adequate healthcare [23]. Finally, LICs were defined as countries with a gross national income per capita lower than 1025 USD in 2018, as per the World Bank guidance [24].

### 2.3. Search Strategy

Based on the research question *What are the barriers women face to accessing maternal care in Low-Income Countries in Africa?*, we identified the following three key concepts: “Maternal Care”, “Access”, and “Low-Income countries in Africa”. Synonyms for the main concepts were used as keywords and subject headings in the selected databases. The list of LICs in Africa was extracted from the World Bank database for fiscal year 2019 [24]. Search concepts, relative synonyms, and subject headings are presented in Table 1.

Various publications were searched for, including systematic reviews, qualitative and mixed-method studies reporting barriers to accessing maternal care by service users and providers. The general search strategy can be found in Appendix A and the search strategy for each of the databases can be found in Appendix A.

### 2.4. Data Collection Process

Database search results were extracted on the 18 May 2019 and exported to Mendeley. Duplicates of the merged database results were eliminated. Initial screening was done based on article titles and abstracts to check for inclusion eligibility. Further in-depth screening was then done to identify articles meeting eligibility criteria by reviewing the methodology sections for each publication. Following this screening process, the Preferred Reporting Items for Systematic Reviews and Meta-Analyses (PRISMA) flowchart was adapted to assort resulted articles into identified papers, screened publications, and included and excluded articles according to the project’s specific criteria [25].

### 2.5. Risk of Bias

We used the Mixed Methods Appraisal Tool (MMAT) to select articles with at least 80% quality score (see Appendix A [26]. Quality assessments were performed independently by the two authors.

## 3. Results

Following the search and elimination of ineligible articles, 21 results were identified. 13 high-quality results were selected based on MMAT scores (Figure 1). The included studies were from eight countries (Table 2). We used thematic analysis to synthesize the study findings that were extracted from the results and discussion sections of the selected articles [27]. Those sections were thoroughly read and analysed.

The detailed study findings are summarized in Table 3.

### 3.1. Thematic Synthesis

The findings were synthesised in themes, revealing a variety of accessibility barriers (Table 4). Among the most common themes were the economic challenges to accessing maternal healthcare, transportation-related issues, and cultural beliefs.

#### 3.1.1. Transportation

Transportation-related barriers were reported in most of the included studies [28,29,30,34,35,37,38,39,40]. Cattle camp residents and pastoralists from South Sudan, Ethiopia, and Mali reported distance as an important barrier to accessing health facilities. Farming and raising livestock are common activities in these countries, where farmers and pastoralists often live in continuous movement [28,35,37,40]. Included studies showed that nomadic women were challenged by the long distances they needed to travel for their antenatal care visit or to deliver in a health facility [35,40]. Furthermore, the absence of a reliable mode of transportation can be an obstacle to reaching a health facility [28,29,30,34,35,37,38,39,40].

Weather further complicated transportation is some countries. Rainy seasons in South Sudan, Togo, Mozambique, and Zimbabwe [28,29,30,33,34,35] can cause roads to flood and evidence showed that this hindered women from seeking facility-based maternal care. In some settings like Togo, walking was the only available method to reach a health facility due to the condition of the roads [29].

Insecurity was another reported barrier to accessing maternal healthcare services, especially in conflict areas, like in South Sudan, for example [28,35,39]. People living in fragile settings were in constant fear and displacement, searching for safety. Reaching health facilities was almost impossible as they could be targeted and be kidnapped or killed [28,35,39].

Finally, unreliable ambulance service can complicate access to maternal care. Pregnant women, their families, and healthcare workers reported that ambulance telephone services could be out of service, even in case of emergencies [29]. A participant from a study in Ethiopia [32] reported:


*You call the ambulance but drivers rarely pick their phone. Even if they pick their phone, they do not come as immediately as you would like them to. Meanwhile, the women may deliver at home as labor at times takes shorter and we have so many of such experiences.*


#### 3.1.2. Culture and Beliefs

Traditions and beliefs influenced utilization of maternal healthcare services. In South Sudan and Ethiopia, some women preferred to give birth at home and were reluctant to attend antenatal care visits and maternal health facilities [28,33,37]. In South Sudan, women reported they could give birth to their baby anywhere naturally, without the need for prior preparations. They perceived the use of maternal health facilities to be restricted to complicated pregnancies, as did women from a study in Mozambique [28,30,35]. Meanwhile, in Tanzania, evidence showed high awareness of birth preparedness [38], while in Ethiopia some women were unconvinced about the benefits of antenatal care. One woman from a study from Ethiopia stated:


*Why do you have to visit health facility for getting pregnant? Pregnancy is not a health problem at all. For me and other women alike running to health facility because you are pregnant is not normal although there is continuous push from UHEPs [Urban Health Extension Professionals].*
[32]

Ethiopian women reported to have preferred home delivery, to allow them to participate in commonly observed religious practices after childbirth, like “onur”, a blessing from a religious leader to the newborns [37].

In some countries, women’s decision to utilize institutional delivery was dependent upon their husbands’ approval [38]. Literature suggests that husbands can force their wives to deliver at home as they believe childbirth to be a *natural duty* and to avoid their wives from being exposed in-front of male healthcare providers during childbirth [38].

A study from Mali reported that it was not acceptable for women to go alone to the health facility and they had to be accompanied by their husbands for their security, cultural acceptance, and for covering the financial expenses of their wives’ medical visits [40]. However, this study revealed that most men were busy working and could not accompany their wives to maternal health facilities. Moreover, Muslim women from Mali were not allowed to disobey their husbands and they could not utilize institutional delivery [40].

#### 3.1.3. Family Support

Women from LICs in Africa shared a huge responsibility for taking care of their household, caring for their children, as well as helping their husbands with farming during harvesting seasons, especially in pastoralist populations and cattle camp communities in South Sudan [28]. Evidence suggests that women were unable to leave their homes and domestic responsibilities to travel long distances to health facilities for attending antenatal care visits, delivering the baby, or going to postnatal care visits, especially when there was no other family member who could take care of their other children [28,30,35,37].

#### 3.1.4. Economic Factors

The direct and indirect costs of maternal care were a recurring barrier to maternal healthcare access as reported by women and relevant stakeholders [28,29,30,33,34,35,37,38,39,40]. Medical fees varied from one country to another according to local regulations and policies [28,30,33]. In South Sudan and Mozambique, women complained about the high cost of institutional delivery even though antenatal care was free of charge at the point of use [28,30,33]. It was not clear whether the reported high cost was related to expensive medication or under the table costs for healthcare workers [28,30]. Other countries, like Togo, had high official cost of delivery, as reported by healthcare workers. Families of lower socio-economic levels could not afford institutional delivery, and even those who delivered in a hospital could stay in debt for years due to the high costs of hospital delivery [29]. Cost of transportation to health facilities was another barrier to accessing antenatal care visits and MHFs. Women from Ethiopia and Zimbabwe reported that high transportation costs and being financially dependent on their husbands hindered them from accessing MHFs [32,34].

#### 3.1.5. Quality of Care

Lack of sufficient bed capacity and private birth space in maternity wards was an issue reported in Togo and South Sudan [33,39]. Reports from Ethiopia revealed that this problem, in addition to a lack of clean water supply and food in health facilities, made women reluctant to deliver in an MHF [37]. Lack of well-equipped clinics for childbirth and the absence of necessary medication in some primary healthcare units were also reported by women in South Sudan [33].

The findings show that women complained about long waiting times and lack of sufficient workforce in health facilities [34,39]. There were often very few healthcare workers, or none at all, when women reached an MHF for attending their antenatal care visit or for delivery [37]. Moreover, evidence implies that some healthcare workers lacked the required experience and knowledge to provide adequate healthcare [31].

Lack of workforce is illustrated by the following quote from a participant from a study in Ethiopia [37]:


*When women come to health facilities for ANC after walking long distances on foot, no health professional is available and the health facility is closed. No services are provided because the health facility is always closed. Thus, why would they [women] come to a health facility the next time?*


Quality of care was also related to the attitudes of healthcare workers. Evidence highlighted that some women considered vaginal examination without prior notice or consultation as an invasive and disrespectful act to women’s privacy [28]. Findings revealed women’s complaints about the lack of privacy and discomfort during examination and childbirth, when they were exposed in front of many healthcare workers and other patients [28,31]. Women also reported lack of utilization of MHF due to their discomfort and reluctance to being examined by male healthcare workers [28]. On the other hand, women from Ethiopia reported preference for male healthcare workers as they considered them to be more professional and empathetic compared to female providers [31].

Verbal and physical abuse during childbirth was another form of disrespect and abuse that was highlighted in some studies [31,32,36]. Some women reported being insulted and even slapped on their legs during childbirth [31]. In Malawi, disrespect and abuse have been reported by women attending both public and private hospitals [36]. Evidence revealed that poor women attending public hospitals were more likely to suffer from verbal and physical abuse [36].

Poor communication barriers can also lead to poor quality of care, especially in multilingual societies, like Ethiopia and Malawi where healthcare workers might not be speaking the same language as patients [31,36]. This can lead to miscommunication and discomfort between women and healthcare workers [31]. There is some evidence that speaking the same language could contribute to a positive relationship between pregnant women and healthcare workers, while the presence of speech impairment was also found to complicate communication [36].

## 4. Discussion

This study contributes to the body of evidence on perceived barriers to accessing maternal care in LICs in Africa. Previous publications suggest that lack of proper access to maternal healthcare can greatly influence maternal death rates [41]. This review provides a comprehensive overview of barriers to accessing maternal healthcare services in LICs in Africa, from the perspective of women and relevant stakeholders [1,5,12,24].

Although some LICs have initiated public health interventions to address maternal mortality, they still under-estimate the need to invest in women’s health [42]. Findings from this review suggest that women in LICs are highly vulnerable as they suffer from poverty, lack of awareness of maternal healthcare benefits, transportation-related barriers to health facilities, and lack of autonomy, in addition to security concerns which prevent them from accessing maternal healthcare facilities, in agreement with previous research [40,43,44,45]. The various barriers are inter-related and complex in nature. A main finding in this review is the importance of economic factors [28,29,30,33,34,35,37,38,39,40] and the uneven provision of healthcare among the different socio-economic levels [46]. While LICs have more than 90% of the global burden of disease, they also have the least access to healthcare, with minimal budget allocations to health [47]. Out-of-pocket spending on healthcare in LICs is enormous and exerting tremendous burden on poor, vulnerable populations, and could lead to economic distress especially in crisis settings [48]. This is in line with the World Bank data from the health financing report and other published literature [23,47,49] where out-of-pocket payments in Sub-Saharan Africa account for more than 40% of the total health expenditure despite the global efforts towards the reduction of such payments [49].

Poverty, low socio-economic status, and low levels of education [28,29,33,36,37] in LICs in Africa intersect and can affect the communities’ awareness of the health benefits of maternal healthcare and institutional delivery [28,29,30,32,33,34,35,37,39,43]. A study from Ghana showed consistent results regarding the lack of women’s and families’ awareness of the benefits of antenatal care visits despite their knowledge of antenatal care availability [44,45]. There is a wide perception that early antenatal care visits are unnecessary as pregnancy is not yet visible [29,30,44], and that such visits could expose women to the evil eye [45] or to shame for revealing pregnancy during the first trimester. Furthermore, most of the reported findings are drawn from rural, poor or distant villages where residents are of lower socio-economic and educational levels and more prone to lack of adequate resources [26,27,28,31,32,33,35,36,37,38]. Families in LICs, especially pastoralists and nomads, strive for survival, and accessing healthcare facilities is a continuous challenge [24,28,35,37,40].

Another finding in this review that needs to be highlighted are the attitudinal barriers women often face when they seek maternal healthcare. Single pregnant women, for example, can be stigmatized, preventing them from accessing MHFs [31,32]. A study from Tunisia revealed discrimination, stigmatization, and abusive treatment of single mothers during childbirth at health institutions [50]. Most women complaining about disrespect and abuse during childbirth in this literature review were from poor socio-economic levels [28,32,36].

This review also showed the importance of infrastructure for maternal healthcare. Pregnant women report poor infrastructure of health facilities, prolonged waiting times, and lack of clean water supply, experienced healthcare professionals, and sufficient bed capacity. Even when women need to utilize MHFs, they are reluctant to go due to the bad reputation of MHFs and the perceived poor QoC [28,29,31,33,34,35,37,39]. These concerns are in line with study results from Uganda that demonstrated a lack of utilization of MHFs due to women’s perception of low quality of care [51]. The findings agree with those of Sibiya et al. that women do not carry on with antenatal care visits if their first experience was bad and of low quality [52].

Insecurity and political conflicts can also lead to significant barriers to accessing maternal healthcare [28,35,39]. In South Sudan, women feared accessing health facilities due to unsafe roads and the threat of attack [28,35] of health facilities and kidnapping [39]. This reflects Red Cross reporting on violent incidents of attacks in conflict areas [53,54]. Around 60% of maternal deaths worldwide occur in fragile settings and conflict zones [13] where access to healthcare is a persisting challenge [28,35,39,55]. Insecurity can stop people from accessing health facilities due to fear of attacks.

Due to conflicts worldwide, 95% of the displaced refugees live in developing countries, which exerts further financial and healthcare burden on LICs [56]. In spite of the ambitious SDG to 2030 that aim for equitable health access and reduction of global maternal mortality [56,57], the world is expected to have increased numbers of poor people living in conflict settings by 2030 [58], which might further compromise access to maternal healthcare.

## 5. Strengths and Limitations

This review has several limitations that need to be considered. We only included published and peer-reviewed sources; it is likely that several of the barriers faced by women in LICs in Africa are recorded through other means, such as internal publications of international organisations. Furthermore, while there are 26 LICs in Africa, the articles included in this review came from only eight of them, with South Sudan and Ethiopia accounting for seven of the thirteen studies that were included, and hence we cannot know whether the results are relevant to all LICs in Africa, especially since African countries show great variability in cultures, religions, and infrastructure. This project did not include barriers related to postnatal care, which is crucial to prevent maternal and neonatal health complications [5]. However, this was beyond the scope of this review, which focused on barriers to accessing maternal health services during pregnancy, childbirth, and immediately postpartum. The inclusion of only observational studies (qualitative and mixed-method design) introduced a methodological bias, but such a choice was necessary because we wanted to explore the barriers reported by women and the effects these had on them.

This literature review is among the first to investigate maternal healthcare access in LICs in Africa following the initiation of the UN SDGs to 2030. We only included studies that scored at least 80% on MMAT to ensure included studies were methodologically robust and we can be moderately confident in the overall results, especially since all studies reported similar barriers. The inclusion of studies focusing on the perspectives of women and other relevant stakeholders allowed an in-depth exploration of a variety of barriers perceived by both maternal healthcare users and providers [59]. This review has explored a variety of barriers to accessing maternal healthcare which are worth flagging to policymakers to assist in setting effective cross-sectoral collaborations towards the SDGs to 2030 [58], aiming to reduce the global maternal mortality ratio to less than 70 per 100,000 live births and to provide universal health coverage and equitable accessibility to healthcare [9].

## 6. Recommendations

Further research on maternal healthcare access is required in LICs [13]. Creating accurate healthcare system registries and databases is crucial for credible documentation and accurate monitoring and evaluation of equitable access to maternal healthcare among vulnerable populations [5]. As access to maternal healthcare is still a challenge for many women in LICs in Africa, and due to the complexity and multi-factorial issues leading to non-utilization of maternal healthcare services, governments, stakeholders, and policymakers need to collaborate to resolve accessibility issues [13]. Among the different level recommendations are the following:

### 6.1. Community-Level

As the Ottawa Charter 1986 has advocated for community engagement to improve community health [60], public health campaigns, with the support of community leaders, need to inform pregnant women about maternal healthcare benefits and support them to utilize MHFs [13,30]. Women’s autonomy also needs to be endorsed and women should be empowered to make their own health decisions through dedicated health education and promotion programmes, targeting pregnant women, their families, and the whole community [5,13].

### 6.2. Governmental Level

Ministries of finance should collaborate with ministries of health to apply special financing programs, such as conditional cash transfer and free maternal healthcare, which have shown improvements in access and utilization of health services in some countries [5]. Reimbursement of transportation cost and cost of stay in health facilities is another suggested intervention that could contribute to overcoming financial barriers and increase accessibility and utilization of health facilities. Adequate budget allocation to maternal healthcare, equipping healthcare units and centers with needed medical supplies, experienced workforce, adequate infrastructure, and ensuring full geographical coverage with MHFs [5,13] and rolling out insurance [61,62], can also lead to increased access to maternal health.

### 6.3. Health Systems

Health professionals and HCWs should be empowered with appropriate knowledge and rigorous ethical and medical training to ensure provision of equitable, high quality care [5,13]. Finally, as this review has illustrated critical barriers relevant to conflict zones like South Sudan [28,35,39], it is crucial to ensure political stability and security to improve maternal health access. As per the Global Strategy for Women’s, Children’s and Adolescent’s Health 2016–2030 [13], there should be cross-disciplinary partnerships to establish strong health systems that safeguard the equitable accessibility and utilization of health services.

## 7. Conclusions

Global action plans are in continuous development to ensure that equitable access to maternal healthcare is achieved by 2030, in line with the Global Strategy for Women’s, Children’s and Adolescent’s Health and the UN SDGs [13]. This systematic review contributes to increasing the body of evidence on barriers to maternal healthcare access in LICs in Africa, focusing on the period following the MDGs for 2015. The results from this review are relevant to SDGs goals 3, 5 and 10, which target proper health and well-being all over the world, empowering women and endorsing their autonomy and roles in the society, as well as providing equitable healthcare access to marginalized and poor populations. Evidence from this review shows that in LICs, cost of transportation and health services, distance, attitudinal barriers, and cultural beliefs are among the most prevailing barriers to maternal healthcare [28,29,30,31,32,33,34,35,36,37,38,39,40]. Maternal health investments should be implemented based on concrete evidence and accurate monitoring and evaluation plans. Further in-depth research is required to monitor and evaluate barriers to accessing maternal healthcare in various low-income settings.

## Figures and Tables

**Figure 1 ijerph-17-04292-f001:**
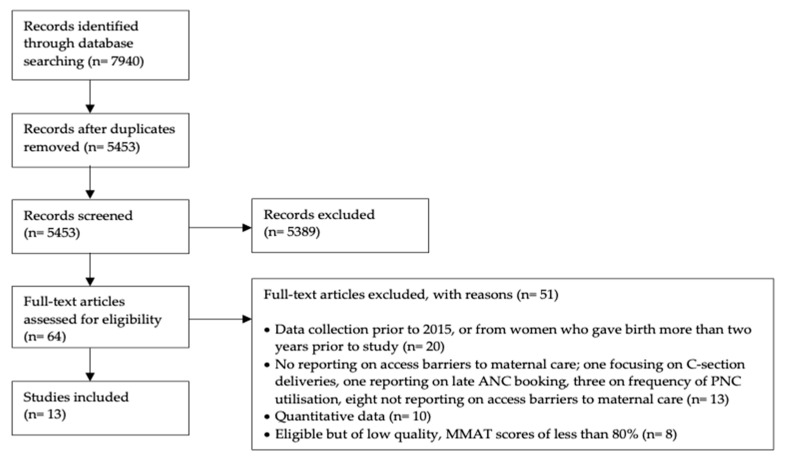
Prisma Flowchart.

**Table 1 ijerph-17-04292-t001:** Search Keywords and Concepts.

Keywords			
Maternal Care	Access	Low-Income Countries in Africa	
		West or Central or East or South Africa Low Income Countries	
			Madagascar
maternal	access	Benin	Malawi
antenatal	barrier *	Burkina Faso	Mali
perinatal	difficult *	Burundi	Mozambique
obstetric	challenge *	Central African Republic	Niger
postnatal	obstacle *	Chad	Rwanda
prenatal	problem *	Comoros	Senegal
ANC	socio *	Congo	Sierra Leone
pregnancy	transport *	Eritrea	Somalia
	stigma	Ethiopia	South Sudan
	culture *	Gambia	Tanzania
	equit *	Guinea	Togo
		Guinea Bissau	Uganda
		Liberia	Zimbabwe

*Note*: We used the symbol “*” to truncate search terms in order to include variations that started with the same letters.

**Table 2 ijerph-17-04292-t002:** Country Classification per publication.

Country	Number of Publications
South Sudan	4
Ethiopia	3
Mali	1
Togo	1
Mozambique	1
Malawi	1
Tanzania	1
Zimbabwe	1

**Table 3 ijerph-17-04292-t003:** Study Findings.

Reference	Author/Year	Country	Aims and Objective	Population	Methods	Findings
[28]	Wilunda et al., 2016	South Sudan	To identify utilization barriers to institutional delivery services in Rumbek North County.	Women and other relevant stakeholders (97% no formal education	Qualitative Focus group discussions (FGDs) with 169 women	1. Barriers to reaching care facilities 2. Insecurity due to conflict 3. Preference for home delivery 4. Poor QoC
[29]	Arnold et al., 2016	Togo	To investigate perceived barriers to accessing reproductive and maternal healthcare services in Kozah district, Northern Togo, focusing on gender-related issues.	Peri-urban and rural catchment areas (majority with no formal education or primary education only)	Qualitative 12 Semi-structured interviews; 4 FGDs and key informant interviews (KIIs) with community leaders and women	*Prenatal* 1. Barriers: Distance related barriers and cost, lack of proper QoC *Delivery* 2. Financial burden, transportation barriers 3. Women’s perception that they need to travel to distant health facilities for better QoC 4. Long waiting times 5. Lack of consideration of women’s concerns 6. Health providers’ respectful attitude towards pregnant women as a facilitator *Postnatal* 7. Cost and lack of proper knowledge of postnatal care
[30]	Munguambe et al., 2016	Mozambique	To understand antenatal care seeking behaviour of women, focusing on structural and socio-cultural barriers to access antenatal care facilities in Gaza Provinces and Maputo, southern Mozambique.	Rural communities in Mozambique 196 participants: women and family members (24% with no formal education, 42% with primary education), Traditional healers, elders, and district medical officers.	Qualitative 33 FGDs with 196 participants and 13 in-depth interviews	1. Health seeking behaviour in case of complications 2. Transportation barriers 3. Lack of birth preparedness 4. Financial barriers of the family 5. Lack of awareness of pregnancy complications 6. Belief of pregnancy complication as a misfortune 7. Women with other children found it difficult to leave their kids and go to the hospital 8. Women showed awareness of antenatal benefits, but attended first visit after first trimester 9. Community belief in the health benefits of institutional delivery 10. Health providers’ mistreatment during childbirth
[31]	Burrowes et al., 2017	Ethiopia	1. Examine women’s experiences of respectful care during labour and childbirth 2. Investigate midwives’ knowledge of patients’ rights, ethical and respectful care 3. Report midwives’ experiences of patient mistreatment 4. Proposed recommendations by patient and midwife to improve QoC.	Urban population, mostly educated 4 midwives, 15 midwifery students, and 23 women who gave birth during the previous year	Cross-sectional, qualitative 45 in-depth interviews	1. Both women and midwives reported verbal abuse 2. Women didn’t report physical abuse, but HCWs did 3. Women not allowed to deliver in their preferred position 4. Non-consented care and abandonment 5. Lack of privacy and minimal bed capacity 6. Directive counselling and unneeded procedures 7. Lack of trained HCWs on patient rights and ethics, but high awareness of patient respectful care 8. Stigmatization for unmarried women 9. Poor QoC and frustration of HCWs due to work stress 10. Communication and language barriers
[32]	Kaba et al., 2017	Ethiopia	To investigate reasons for non-use of available maternal healthcare services in urban areas in Ethiopia.	Community members from urban slums (low educational levels)	Qualitative, explorative 40 in-depth interviews and 11 FGDs	1. Vulnerable women like migrants and women living in slums didn’t utilize MHFs 2. Cultural beliefs and religion; shame when pregnancy occurs to single women 3. Working pregnant women requiring employer’s approval to leave work for medical appointments 4. Lack of knowledge about benefits of maternal healthcare 5. Lack of ambulance service 6. Disrespect and abuse by HCWs discouraged pregnant women to deliver in MHFs 7. Lack of financial capabilities for frequent antenatal care visits and delivering in a health facility
[33]	Lawry et al., 2017	South Sudan	To investigate barriers to maternal, newborn and child healthcare in under-developed state in South Sudan.	860 pregnant women or mothers with children under 5 years old, 144 men (more than 90% no formal education)	Mixed method Quantitative surveys with 860, 72 KIIs with stakeholders, and 25 interviews with community members	Barriers to maternal healthcare: 1. Lack of proper infrastructure and medical equipment 2. Preference for home delivery 3. Distance to health facilities and high charged fees 4. Decision and authority of mother in law 5. Cost 6. Distance barriers
[34]	Nyathi et al., 2017	Zimbabwe	To explore influences of antenatal care service utilization among women in Mangwe district.	15 mothers to babies younger than 12 months living in distant, rural areas	Qualitative, explorative Semi-structured interviews	1. Transportation related barriers 2. Cost 3. Poor quality of care, and disrespect and abuse by HCWs
[35]	Wilunda et al., 2017	South Sudan	To explore barriers to antenatal care service utilization in Rumbek North.	169 women and 45 men from villages and cattle camps (96.7% of women with no formal education) Community leaders	Qualitative FGDs and KIIs	1. Transportation related barriers 2. Socio-Cultural factors 3. Lack of awareness of pregnancy health and antenatal care benefits
[36]	Madula et al., 2018	Malawi	To examine the nature of communication between HCWs and patients in maternity wards and how this affects maternal healthcare.	Women admitted for childbirth in urban hospitals (around 70% educated)	Qualitative, descriptive in-depth interviews	1. Verbal abuse and disrespect reported by few women 2. Failure of HCWs to answer patients’ questions 3. Discriminatory treatment based on socio-economic status in public hospitals, but not private hospitals 4. Linguistic barriers and lack of knowledge of sign language by HCWs
[37]	Medhanyie et al., 2018	Ethiopia	To investigate utilization barriers of reproductive, maternal, and neonatal health services among pastoralist women of Afar, Ethiopia.	5 women, 5 men, community leaders from pastoralist community in Afar (majority of women with no formal education)	Qualitative, explorative 10 FGDs and 45 KIIs	Maternal healthcare services are mostly accessed by wealthy, educated women, living in urban areas Barriers: 1. Lack of awareness of postnatal care visits and benefits of institutional delivery 2. Cultural influence on accessing MHFs 3. Transportation related barriers and cost of care 4. Inadequate resources in MHFs, poor QoC and infrastructure.
[38]	Kohi et al., 2018	Tanzania	To investigate the behaviour of women and men regarding the timing when pregnant women go to hospitals, the place of delivery and the decision-makers in the process of delivery and accessing health facilities for childbirth in Tanzania.	23 participants: women (aged between 20–45) recently giving birth and men (aged between 25–60) attending postnatal clinics from Lake Zone (most participants with primary education)	Part of a large community project: qualitative explorative 4 FGDs and 12 semi-structured interviews	1. Distance related barriers 2. Women and husbands prefer institutional delivery due to better QoC 3. Lack of QoC and respect in public hospitals 4. Preference of private institutional deliveries but high cost prevented their utilization
[39]	Mugo et al., 2018	South Sudan	To explore the experience and perceptions of women and men utilizing maternal healthcare services and the relevant barriers in South Sudan.	30 mothers with babies younger than 3 months, 15 men from Juba town (around 63% of women no formal education)	Qualitative, explorative and descriptive In-depth interviews	1. Geographic accessibility 2. Unaffordable user fees for antenatal care services 3. Low QoC 4. Security/cultural preference for childbirth 5. Availability of adequate health facilities and resources
[40]	Ahmed et al., 2018	Mali	To investigate the socio-cultural influences on assisted childbirth by nomadic women.	Nomadic women who gave birth 3 months before data collection	Qualitative Semi-structured interviews	1. Women gave birth on their own at home, but showed interest in institutional delivery due to better QoC 2. Women seeking MHFs in case of pregnancy complications 3. Transportation on damaged roads required women to be accompanied by men to health facility

**Table 4 ijerph-17-04292-t004:** Themes.

Themes	Barriers
Transportation	Distance [28,29,30,32,33,34,37,38,39] Weather and poor roads [28,29,30,33,34,40] Security [28,35,39] Lack of ambulance service [30,32]
Culture and beliefs	Traditions and beliefs [28,30,32,33,35,37,39] Lack of belief in or limited knowledge of benefits of maternal healthcare [28,29,30,32,33,34,35,39] Lack of women’s autonomy [28,31,33,35,38,40]
Family support	Domestic chores and family obligations [28,30,35,37] Lack of husband’s support [28,30,32,33,37,39,40]
Economic factors	Cost of maternal care and transportation [28,29,30,33,34,35,37,38,39,40]
Quality of care	Lack of proper infrastructure [28,29,33,37] Insufficiently experienced providers [31,34,35,37,39] Disrespect and abuse [28,29,31,32,34,36] Language barriers [36]

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
