# Peer review of "Barriers to Accessing Maternal Care in Low Income Countries in Africa: A Systematic Review"

_ijerph, 2020, doi:10.3390/ijerph17124292_

Round 1

Reviewer 1 Report

Good job on the manuscript. Here are some suggestions that I think can further improvise the manuscript:

The sentence ‘At least a 10-fold return on women’s, 37 new-borns’ and children’s health will be achieved through better social contribution and increased 38 productivity, in addition to better educational attainment’ in the background seems to break the flow and is hard to interpret. Initially, you mention that economies improved with reduced mortality. However, the following sentence does not appear to be a continuing statement. I would just avoid thee sentence as it is not quite matching the scope of the study.

I would avoid abbreviating Sub-Saharan Africa. It doesn’t appear too many times.

Please keep the objectives as the final sentence of the background. The explanation of objectives after stating them before methods break the flow.

Methods

Did you check the database for new articles since your search ended in May 2019? Mention of any new studies that were not included in the study in the discussion section would be helpful.

I did not see any section on the overall quality of evidence of selected studies. Please add this. Refer to the GRADE approach for further details. It would be nice to grade the overall quality of evidence that you present in order to gauge the applicability of results.

Author Response

Thank you very much for these very useful comments, we really appreciate them.

Please find our responses below:

1. The sentence ‘At least a 10-fold return on women’s, 37 new-borns’ and children’s health will be achieved through better social contribution and increased 38 productivity, in addition to better educational attainment’ in the background seems to break the flow and is hard to interpret. Initially, you mention that economies improved with reduced mortality. However, the following sentence does not appear to be a continuing statement. I would just avoid thee sentence as it is not quite matching the scope of the study.

ANSWER: We removed this sentence and also edited the preceding sentence.

2. I would avoid abbreviating Sub-Saharan Africa. It doesn’t appear too many times.

ANSWER: We have eliminated the abbreviation and use Sub-Saharan Africa in full.

3. Please keep the objectives as the final sentence of the background. The explanation of objectives after stating them before methods break the flow.

ANSWER: The background now finishes with the objectives.

Methods

4. Did you check the database for new articles since your search ended in May 2019? Mention of any new studies that were not included in the study in the discussion section would be helpful.

ANSWER: We did not re-run the search and therefore the discussion did not include any new studies. There were some relevant new studies in the discussion, but these were studies that did not meet the criteria to be included in the review.

5. I did not see any section on the overall quality of evidence of selected studies. Please add this. Refer to the GRADE approach for further details. It would be nice to grade the overall quality of evidence that you present in order to gauge the applicability of results.

ANSWER: Thank you for this very useful comment. We added a statement about the overall quality of evidence on page 15, lines 319-322. We are, however, reluctant to make reference to the GRADE framework because we did not do the extensive evaluation that this GRADE suggests.

Reviewer 2 Report

The study is very interesting, although the review does not contemplate quantitative aspects, the qualitative analysis, despite the limitations (countries analyzed, for their contribution to the literature, generally South Sudan and Ethiopia) makes a description of the situation of barriers to access to maternal care in SCI very detailed and interesting.It deserves to be published.

I suggest to shorten table 3 length

Author Response

Thank you very much for these very useful comments, we really appreciate them.

Please find our responses below:

1. I suggest to shorten table 3 length

ANSWER: Prior to submission we sought to make this table as short as possible. We carefully reviewed the table, but we think it is important for the information to remain so that the reader can form a comprehensive picture of the available evidence.

Reviewer 3 Report

Thank you for the possibility to review the article “Barriers to Accessing Maternal Care in Low Income 2 Countries in Africa: A Systematic Review”. The manuscript is within the scope of the journal.

The authors conducted a systematic meta-analysis on the topic of access to antenatal care in low income countries in Africa. The introduction is well written and conclusive. The methodology of the research is clearly described. Tables are conclusive, but should be again controlled for line break.

The section “results” includes aspects of discussion. Still it is informative and clearly written.

The discussion is very well structured. The strengths and weaknesses of the study are discussed, especially considering the selection of countries of which studies are identified. The recommendations for further research are well presented and offer the opportunity to use this study as the basis for further studies and political statements.

Minor mistakes: Table 4 T-ransportation

Author Response

Thank you very much for these very useful comments, we really appreciate them.

Please find our responses below:

1, The section “results” includes aspects of discussion. Still it is informative and clearly written.

ANSWER: We reviewed the section carefully and ensured it reports the findings.

2. Minor mistakes: Table 4 T-ransportation

ANSWER: We corrected this